# Natural Flavones and Flavonols: Relationships among Antioxidant Activity, Glycation, and Metalloproteinase Inhibition

**Simone Ronsisvalle, Federica Panarello, Giusy Longhitano, Edy Angela Siciliano, Lucia Montenegro * and Annamaria Panico**

Department of Drug Sciences, University of Catania, Viale A. Doria 6, 95125 Catania, Italy;
s.ronsisvalle@unict.it (S.R.); panafede@hotmail.it (F.P.); longhitano.giusy87@tiscali.it (G.L.);
edysiciliano@hotmail.it (E.A.S.); panico@unict.it (A.P.)
* Correspondence: lmontene@unict.it; Tel.: +39-095-738-4010

**Abstract:** Reactive oxygen and nitrogen species as well as advanced glycation endproducts (AGEs) and metalloproteinases (MMPs) play a key role in the development and progression of degenerative processes of body tissues, including skin. Natural antioxidant flavonoids could be beneficial in inhibiting AGEs' formation and MMPs' expression. In this study, the antioxidant activity of flavones (luteolin, apigenin, and chrysin) and flavonols (mirycetin, quercetin, and kaempferol) was compared with their inhibitory effects on both metalloproteinases' (MMP-1, MMP-2, MMP-9, MMP-13) and AGEs' formation. Comparisons were performed taking into account the hydroxyl group arrangement and the physico-chemical parameters the binding dissociation enthalpy (BDE), ionization potential (IP), partition coefficient (log P), and topological polar surface area (TPSA). Increasing the number of hydroxyl groups led to a proportional enhancement of antioxidant activity while an inverse relationship was observed plotting the antioxidant activity vs. BDE and IP values. All flavonoids acted as AGEs, MMP-1, and MMP-13 inhibitors, but they were less effective against MMP-2 and MMP-9. The inhibition of MMP-1 seemed to be related to the TPSA values while high TPSA and low log P values seemed important conditions for inhibiting MMP-13. Overall, our data suggest that an estimation of flavonoid activity could be anticipated based on their physico-chemical parameters.

**Keywords:** antioxidants; polyphenols; anti-glycation activity; metalloproteinases; Maillard reaction

## 1. Introduction

The Maillard reaction, also known as glycation, is a chemical reaction between amino acids and reducing sugars. The reaction can be divided into two parts. The first half proceeds by the rearrangement of Amadori, and the second half leads to the formation of advanced glycation endproducts (AGEs). These processes involve various reactions, such as oxidation, dehydration, and condensation. Glycation, which includes oxidative phases [1], can induce direct cell damage [2], promoting various diseases, such as tissue damage, arthrosis, cancer, diabetes, and cutaneous disorders, including wrinkle formation and sagging skin [3,4]. In the oxidative glyco-process, the undue production of reactive oxygen and nitrogen species (ROS and RNS) increases the non-enzymatic glycation reaction resulting in AGEs' formation [5]. AGEs induce the release of inflammatory and degenerative factors, such as interleukin-1$\beta$, tumor necrosis factor (TNF)-$\alpha$ [6], and metalloproteinases (MMPs) [7]. The activation of MMPs causes progressive damage to the components of the extracellular matrix (ECM), in particular the degradation of proteoglycans (PGs) and the destruction of collagen, with consequent tissue structural modifications [8–10]. Several

natural antioxidant polyphenols have been harnessed in the treatment of disorders involving glyco/oxidative processes [11,12]. The intake of natural antioxidant polyphenols strengthens the defenses against the glico/oxidative process, attenuating endothelial dysfunctions related to the inhibition of MMPs [13,14]. Their anti-glycation/antioxidant ability can be partly related to the chemical structure. The arrangement of hydroxyl side groups [15] affects their free radical scavenger activity, depending on the properties of the dissociation energies of the bonds [16]. In fact, the antioxidant capacity of polyphenols depends on their ability to donate hydrogen and transfer the electron [17], correlated with the enthalpy values of the binding dissociation enthalpy (BDE) and ionization potential (IP), respectively [18]. In addition, the planar conformation and the geometry of the molecule can affect the resulting antioxidant ability, providing information on the rate of radical formation [19]. Several researches have investigated the correlations between the chemical structure and activity of flavonoids. For flavonoids to exert their antioxidant and radical scavenging activity, the importance of having an unsaturated 2-3 bond in conjugation with a 4-oxo function in the flavone skeleton and the presence and position of hydroxyl groups in the ring B has been widely reported [20–24]. Eto et al. [25], analyzing the relationship between the radical-scavenging ability of some flavonoids and their BDE and IP values, found a correlation with BDE but not with IP. Investigating flavonoids' inhibitory ability against protein glycation, Wu et al. [26] observed a good relationship with their free radicals scavenging effect that made the authors attribute the anti-glycation activity of flavonoids to their antioxidant properties. A similar correlation between anti-glycation and antioxidant activity was reported for flavonoids contained in tropical medicinal herbs [27]. Sim et al. [28] pointed out that the inhibitory effects of flavonoids on MMP-1 expression was related to their antioxidant properties and was dependent on both the number of OH groups in their structure and the presence of a C-3-hydroxyl group since flavones were weaker inhibitors than flavonols. On the contrary, the presence of a hydroxyl group at the 3 position was less important in determining the inhibitory effect on MMP-2 and MMP-9 [29]. In addition, drug absorption and interaction with MMPs may be affected by compound lipophilicity, expressed as the octanol-water partition coefficient (log P), and topological polar surface area (TPSA), resulting from the sum of the surfaces of polar atoms in a molecule [30].

To date, studies have been focused on the correlation between the antioxidant effect of several flavonoids and a single expression of their potential biological activity. Therefore, there is a lack of information about the ability of flavonoids with different physico-chemical and antioxidant properties to affect simultaneously the glycation process and MMPs formation. Such information could be helpful to choose the most suitable flavonoids in the treatment of skin disorders involving glycoxidation processes and AGEs formation, such as photo- and chrono-ageing.

Therefore, in the present study, the free radical (ROO• and NO•) scavenger effects of flavones (chrysin, apigenin, luteolin) and flavonols (mirycetin, quercetin, and kaempferol) with a different number of hydroxyl groups in the ring B were evaluated and the resulting data were compared with both in vitro anti-glycation activity and metalloproteinases' inhibition effect. In particular, the inhibition of MMP-1, MMP-13, MMP-2, and MMP-9, the four major matrix metalloproteinases regarded as the most vital enzymes for the degradation of the main constituents of the ECM [8], was evaluated. Furthermore, the correlation among flavones' and flavonols' in vitro activity (antioxidant, anti-glycation, MMPs inhibition activity) and their calculated physico-chemical properties (log P, TPSA, BDE, and IP) was investigated, in an attempt to point out a structure–activity relationship (SAR) that could be useful in the development of cosmetic products aimed at preventing and/or treating skin disorders involving glycation and AGEs' formation.

## 2. Materials and Methods

### 2.1. Materials

Chrysin, apigenin, luteolin quercetin, myricetin, kaempferol, fluorescein (FL), AAPH (2,2' azobis (2-methylpropionamide) dihydrochloride), aminoguanidine carbonate (AMG), 97%, Trolox (6-hydroxy-2,5,7,8-tetramethylchroman-2-carboxylic acid), HEPES (4-(2-Hydroxyethyl) piperazine-1-

ethanesulfonic acid), bovine serum albumin (BSA), D-fructose, sodium nitroprusside Griess reagent (1% dihydrochloride sulphanylamide and 0.1% naphthylethylenediamine (NED) dihydrochloride in 5% of hydrochloric acid), and sodium azide were purchased from Sigma-Aldrich srl (Milan, Italy). OmniMMP Fluorescent Substrate Mca-Pro-Leu-Gly-Leu-Dpa-Ala-Arg-NH$_2$, MMP-9 (refolded) (human) (recombinant) (Catalytic Domain), and MMP-1,-2,-3,-9,-13 (catalytic domain) (human) (recombinant) were purchased from Vinci-Biochem S.r.l. (Firenze, Italy).

### 2.2. Oxygen Radical Absorbance Capacity (Orac) Assay

The ORAC assay was performed as previously reported [31,32]. Briefly, the scavenging effect on peroxyl radical (ROO•), generated by thermo-decomposition of 2,2 azobis (2-aminopropane) dihydrochloride (AAPH) (100 mM), was determined by monitoring the fluorescence decay due to fluorescein oxidation (10 nM) at 37 °C, pH 7.0 using a VICTOR Wallac 1420 Multilabel Counters fluorimeter (PerkinElmer, Waltham, MA, USA) (excitation $\lambda$ = 540 nm, emission $\lambda$ = 570 nm). The control standard and blank were Trolox (12.5 µM) and phosphate buffer, respectively. Samples (100 µg/mL) were properly diluted prior to analysis. ORAC values were calculated as the net protection area under the quenching curve of fluorescein, using Origin®7 (OriginLab Corporation, Northampton, MA, USA). ORAC Units were calculated and expressed as Trolox micromole per microgram of sample (µmol/µg):

$$\text{ORAC value (µmol/µg)} = K(S\ \text{sample} - S\ \text{blank})/(S\ \text{Trolox} - S\ \text{blank}) \tag{1}$$

where K is the sample dilution factor, and S is the area under the fluorescence decay curve of the sample, Trolox, or blank.

### 2.3. NO Scavenger Assay

The percentage of inhibition of spontaneous NO production from an aqueous solution of sodium nitroprusside (20 mM) was determined at 25 °C for 3 h, using Griess reagent, upon the addition of the compound under investigation (1 mg/mL). The reference compound was curcumin (100 µg/mL). Absorbance was measured at 540 nm with a spectrophotometer (Thermo Scientific Multiskan® EX.). The amount of NO radicals was calculated as follows (Equation (2)):

$$\text{\% of inhibition of NO} = [A_0 - A_1]/A_0 \cdot 100 \tag{2}$$

where $A_0$ is the absorbance of the untreated sample, and $A_1$ is the absorbance of the treated samples.

### 2.4. Anti-glycation Activity

The method of Derbrè et al. [33] was used to assess the anti-glycation activity of the investigated compounds. The inhibition of fluorescence produced by AGEs formation through Maillard reaction was determined using a VICTOR Wallac 1420 Multilabel Counters fluorimeter (PerkinElmer, Waltham, MA, USA) ($\lambda$exc 370 nm; $\lambda$em 440 nm).

Protein model bovine serum albumin (BSA) (10 mg/ mL) incubated with D-fructose (0.5 M) in phosphate buffer 50 mM, pH 7.4 and NaN$_3$ 0.02 % was used as a positive control while BSA alone was the negative control. The reference compound was aminoguanidine (AMG) (400 µg/ mL). Final glycated BSA solutions (300 µL) alone and with samples (400 µg/mL) were incubated at 37 °C in 96-well microtiter closed with their silicon lids for 7 days. Results were expressed as relative fluorescence units (RFU) and the percentage of inhibition compared to the positive control (BSA with fructose) was calculated according to Equation (3):

$$\text{\% of inhibition} = \{1 - [\text{RFU sample/RFU Positive control}]\} \times 100 \tag{3}$$

### 2.5. MMP Inhibition Assay

The assay calculates the inhibition of the hydrolysis of the fluorescence-quenched peptide substrate Mca-Pro-Leu-Gly-Leu-Dpa-Ala-Arg-NH due to MMPs' ability to break the Gly-Leu peptide

bond, releasing the fluorophore (Mca) that emits fluorescence at 393 nm after excitation at 328 nm. MMP inhibitors prevent the fluophore release. HEPES buffer (500 mM) containing $CaCl_2$ (5 mM), $ZnCl_2$ (0.1 mM), and Brij-35 (0.05%) was used as medium, at pH 7, in the presence of 10 μM MMPs as proteolytic enzymes, and 2 μM of peptide substrate. After incubation at 25 °C with increasing concentrations (0.0004, 0.004, 0.04, 0.4, 4, 40, 400 μM) of tested compounds, the fluorescence (excitation maximum 328 nm; emission maximum 393 nm) was measured for 3 min after the addition of the substrate using a Victor Wallac 1420 Multilabel Counters fluorimeter (PerkinElmer, Waltham, MA, USA). The fluorescence values of the samples were plotted against time and the regression analysis was performed using the Origin software, version 7 to obtain the slope of the line for each inhibitor concentration. The increase of the slope corresponds to a decrease of inhibitory activity. The $IC_{50}$ values were obtained from a nonlinear regression procedure (log inhibitor vs. response) with a variables slope using the software Origin, version 7. The $IC_{50}$ value was obtained from the equation: $Y = P1/(1 + X/P2)$ [32].

### 2.6. QSAR Analysis

The Quantitative Structure Activity Relationship (QSAR) was calculated with Molecular Operating Environment (MOE) software [16]. In this study, 2-D molecular descriptors, binding dissociation enthalpy (BDE), ionization potential (IP), logP, and topological polar surface area (TPSA) were used. QSAR model evaluation permitted us to correlate the biological activities of these compounds with their physico-chemical properties.

## 3. Results and Discussion

Polyphenols, chemically characterized as compounds with phenolic structural features, are a group of highly diverse natural products and contain several sub-groups of phenolic compounds. Undoubtedly, flavonoids are the most abundant polyphenols in the human diet, representing about 2/3 of all ingested polyphenols. Like other phytochemicals, they are secondary products of plant metabolism. The flavan skeleton of all flavonoids is a 15-carbon phenylpropanoid core (C6-C3-C6 system), arranged into two aromatic rings (A and B) linked by a heterocyclic pyran ring (C) (Figure 1). Generally, the B ring is attached to position 2 of the C ring, but it can also be bound to position 3 or 4. Flavonoids are divided into several groups depending on the oxidation status and saturation of the heterocyclic ring. Therefore, flavonoids include flavones and isoflavones, flavanones, flavonols, 3-deoxy flavonoids, and anthocyanins [1]. Flavones are one of the largest groups, whose main features are a double bond between C-2 and C-3, a ketone group at position 4, and the attachment of the B ring to C-2. Flavones differ from flavonols because of the absence of the hydroxyl moiety in the 3-position (Figure 1).

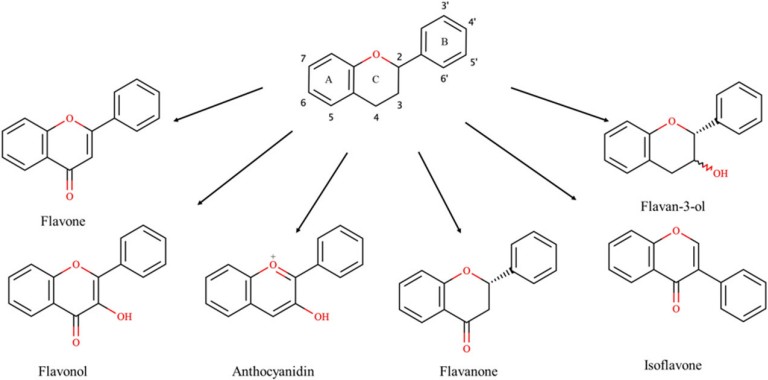

**Figure 1.** Chemical structures of flavonoids.

MMPs' inhibition ability of several flavonoids with radical scavenging properties can be attributed both to their direct interaction with MMPs or to their radical scavenger activity and AGEs'

reduction. In particular, the inhibition of the glycation process can occur by different mechanisms, namely preventing Amadori adduct oxidation, trapping reactive dicarbonyl compounds, and reducing the expression of the receptor for AGEs (RAGE). These mechanisms are supported by some polyphenol features, such as the presence of a C2-C3 double bond and the number and position of hydroxyl, glycosyl, and O-methyl groups. All these characteristics induce changes in the molecular planarity, interfering with the hydrogen bond formation, electron delocalization, and metal ion chelation.

In this work, in the attempt to highlight structure–activity relationships (SARs) in the class of flavones and flavonols, the antioxidant, anti-glycation, and MMP inhibition effects of flavones (chrysin, apigenin, luteolin) and flavonols (mirycetin, quercetin, and kaempferol) were determined and the obtained values were compared with TPSA, log P, BDE, and IP calculated data. All flavonoids contained a double bond between C2 and C3 in their skeleton and they were characterized by a different number of hydroxyl groups: Chrysin has no substitution in the B ring, apigenin has an –OH at the 4′ position, and luteolin possesses two hydroxyl groups at the 3′ and 4′ positions. Quercetin, a major representative of the flavonol subclass, shows two -OH groups at the 3′ and 4′ positions. Kaempferol lacks the -OH group at position 3′ and myricetin has an extra -OH group at position 5′.

The antioxidant and anti-glycation activities of the investigated flavones and flavonols are illustrated in Table 1 and Table 2, respectively. All tested flavones possessed higher ORAC (oxygen radical absorbance capacity) values than the reference compound Trolox, whose ORAC units were equal to 1. In particular, ORAC values decreased in the order luteolin > apigenin > crysin for flavones and myricetin > quercetin > kaempferol for flavonols. These results pointed out a decrease of the peroxyl radical scavenger effect by reducing the number of -OH groups in both subclasses of flavonoids.

**Table 1.** Antioxidant and anti-glycation activity of flavones. Data were expressed as mean ± SD of three determinations.

| FLAVONES | ORAC Units (μM) | NO Scavenger (%) | AGEs Inhibition (%) | MMP-1 ($IC_{50}$) | MMP-13 ($IC_{50}$) | MMP-2 ($IC_{50}$) | MMP-9 ($IC_{50}$) |
|---|---|---|---|---|---|---|---|
| LUTEOLIN | 2.03 ± 0.11 | 48.19 ± 0.18 | 60.04 ± 0.10 | 3.0 ± 0.11 | 1.0 ± 0.14 | 20.0 ± 0.13 | 22.0 ± 0.23 |
| APIGENIN | 1.39 ± 0.13 | 44.81 ± 0.26 | 59.10 ± 0.12 | 6.0 ± 0.13 | 3.0 ± 0.12 | 16.0 ± 0.11 | 18.0 ± 0.15 |
| CHRYSIN | 1.20 ± 0.10 | 35.00 ± 0.28 | 45.08 ± 0.10 | 7.0 ± 0.22 | 10.0 ± 0.24 | 33.0 ± 0.18 | 31.0 ± 0.17 |

**Table 2.** Antioxidant and anti-glycation activity of flavonols. Data were expressed as mean ±SD of three determinations.

| FLAVONOLS | ORAC Units (μM) | NO Scavenger (%) | AGEs inhibition (%) | MMP-1 ($IC_{50}$) | MMP-13 ($IC_{50}$) | MMP-2 ($IC_{50}$) | MMP-9 ($IC_{50}$) |
|---|---|---|---|---|---|---|---|
| MIRYCETIN | 2.89 ± 0.09 | 50.10 ± 0.18 | 58.04 ± 0.11 | 3.0 ± 0.11 | 2.00 ± 0.13 | 17.00 ± 0.23 | 19.00 ± 0.31 |
| QUERCETIN | 2.70 ± 0.10 | 40.01 ± 0.09 | 52.04 ± 0.12 | 1.0 ± 0.16 | 3.00 ± 0.17 | 12.00 ± 0.19 | 22.00 ± 0.16 |
| KAEMPFEROL | 1.40 ± 0.11 | 35.00 ± 0.26 | 38.02 ± 0.17 | 4.0 ± 0.14 | 2.00 ± 0.15 | 24.00 ± 0.21 | 27.00 ± 0.23 |

As shown in Figure 2, the ORAC values and number of hydroxyl groups in the flavone or flavonol molecules were directly related, as increasing the number of -OH groups led to a proportional enhancement of ORAC values ($r^2 = 0.9108$ for flavones and $r^2 = 0.8439$ for flavonols).

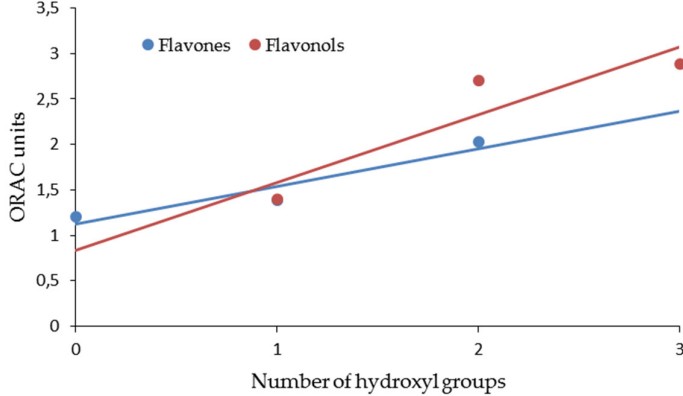

**Figure 2.** Relationships between ORAC (oxygen radical absorbance capacity) values and the number of hydroxyl groups in the flavone and flavonol molecules.

Similar relationships (Figure 3) were observed by plotting the percentage of NO scavenger capacity of tested flavonoids vs. the number of –OH groups in their molecule ($r^2 = 0.9266$ for flavones and $r^2 = 0.9645$ for flavonols).

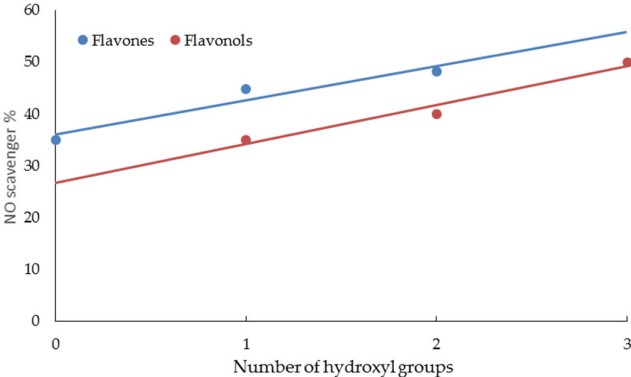

**Figure 3.** Relationship between NO scavenger % and number of hydroxyl groups of flavones and flavonols.

In an attempt to relate the free radical (ROO• and NO•) scavenging activity of flavones and flavonols to the physico-chemical properties that may influence their behavior as a radical scavenger, the BDE and IP of these compounds were taken into account. Such parameters were calculated and are reported in Table 3 and Table 4 for flavones and flavonols, respectively.

**Table 3.** Binding dissociation enthalpy (BDE), ionization potential (IP), topological polar surface area (TPSA), and partition coefficient octanol/water (log P) of flavones.

| FLAVONES | BDE (Kcal • mol⁻¹) | IP (Kcal • mol⁻¹) | TPSA | Log P |
|---|---|---|---|---|
| LUTEOLIN | 74.54 | 174.44 | 107.22 | 2.26 |
| APIGENIN | 82.20 | 176.05 | 86.99 | 2.20 |
| CHRYSIN | 91.85 | 176.00 | 66.76 | 2.50 |

**Table 4.** Binding dissociation enthalpy (BDE), ionization potential (IP), topological polar surface area (TPSA), and partition coefficient octanol/water (log P) of flavonols.

| FLAVONOLS | BDE (Kcal·mol$^{-1}$) | IP (Kcal·mol$^{-1}$) | TPSA | Log P |
|---|---|---|---|---|
| **MIRYCETIN** | 71.08 | 161.40 | 147.68 | 1.76 |
| **QUERCETIN** | 72.35 | 166.08 | 127.45 | 2.03 |
| **KAEMPFEROL** | 83.80 | 167.99 | 107.22 | 2.31 |

Plotting ORAC units vs. BDE (Figure 4) or IP (Figure 5) values for flavones and flavonols, an inverse relationship between these two parameters in both flavonoid subclasses was observed.

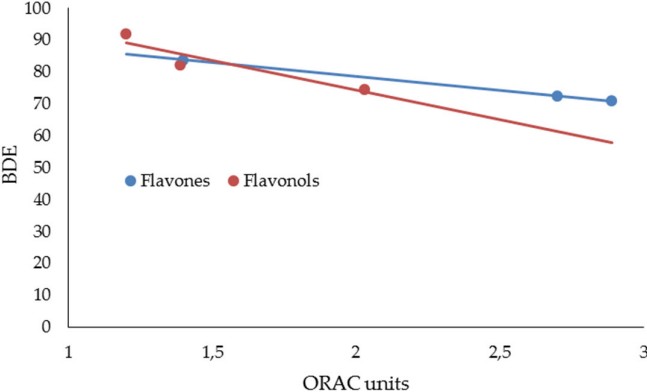

**Figure 4.** Relationships between the oxygen radical absorbance capacity (ORAC) and binding dissociation enthalpy (BDE) values for flavonols and flavones.

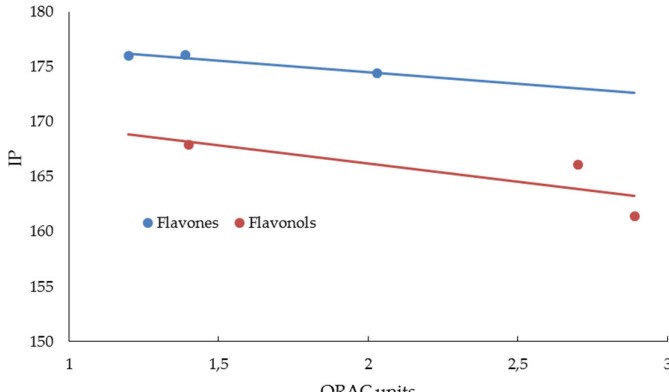

**Figure 5.** Relationships between the oxygen radical absorbance capacity (ORAC) and ionization potential (IP) values for flavonols and flavones.

An analogous inverse relationship between NO scavenger activity and the BDE and IP values of the investigated flavones and flavonols was observed (Figures 6 and 7).

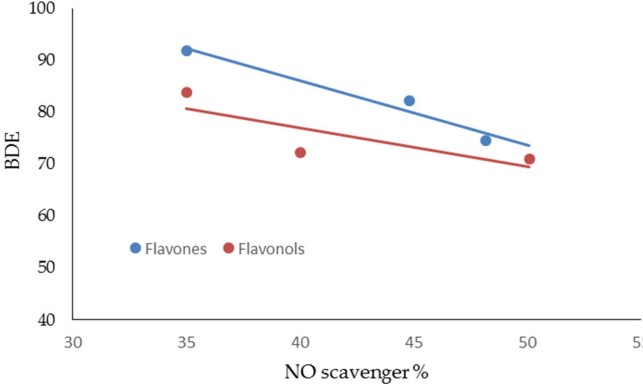

**Figure 6.** Relationships between the NO scavenger % and binding dissociation enthalpy (BDE) values for flavones and flavonols.

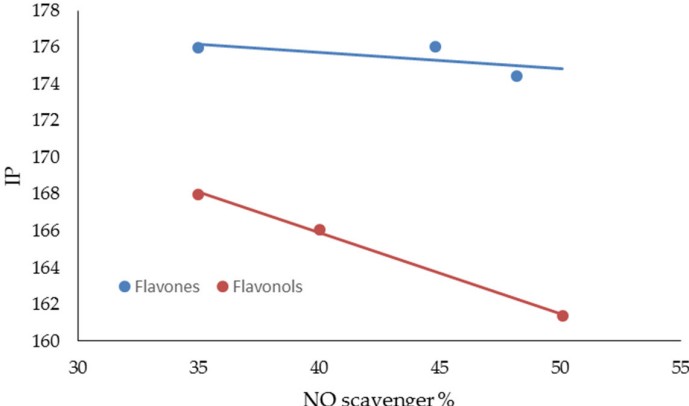

**Figure 7.** Relationships between the NO scavenger % and ionization potential (IP) values for flavones and flavonols.

Therefore, these results pointed out that BDE and IP had a similar effect on the free radical scavenging ability (ROO• and NO•) of the investigated compounds. However, the correlation coefficient ($r^2$) of some of these comparisons was quite low. For instance, by plotting NO scavenger % vs. IP for flavones, we obtained $r^2$ = 0.4602, thus suggesting that a single physico-chemical parameter, such as BDE or IP, could not be regarded as predictive of the antioxidant activity of these compounds, but at least two parameters should be considered to account for the antioxidant effect of the flavonoids under investigation.

Glyco/oxidative processes promote the destruction of the extracellular matrix (ECM), whose complex structure of macromolecules should not be affected to avoid tissue damage. Free radicals lead to direct or indirect tissue alterations [31]. Analogously, the accumulation of AGEs in tissues contributes to the production of free radicals, altering the structure of the matrix and causing an activation of the metalloproteinases. All these factors are responsible for protein fragmentation [34,35]. As reported in the literature, the antioxidant flavonoids can reduce or inhibit AGEs' formation, acting as free radicals scavengers and metal chelators and/or preventing the oxidation of glucose and of Amadori adducts [36,37].

In our study, all tested flavones strongly inhibited AGE production, showing an anti-glycation effect higher than that of the reference compound aminoguanidine whose anti-glycation activity was 40.05 ± 0.32%. In particular, luteolin and apigenin provided a similar AGE inhibitory effect (60.04 ±

0.52% for luteolin and 59.10 ± 0.12% for apigenin) while chrysin showed a lower activity (45.08 ± 0.10%). Therefore, these results suggested that the presence of hydroxyl groups in the B ring played an important role in determining flavones' ability to inhibit AGE production. A similar trend was observed when performing the AGE inhibition assay on flavonols. Although kaempferol had a hydroxyl group in the B ring, it demonstrated an activity lower than that of the reference compound aminoguanidine, thus suggesting that other factors, apart from the presence of hydroxyl groups in the B ring, could be involved.

MMPs are a family of 25 structurally and functionally related enzymes involved in the turnover and degradation of extracellular matrix (ECM) proteins, such as collagen, gelatins, fibronectin, elastin, and laminin. The most known MMPs are classified into collagenases (MMP-1, MMP-8, MMP-13, and MMP-18), gelatinases (MMP-2 and MMP-9), stromelysins (MMP-10, MMP-11, and MMP-27), and matrilysins (MMP-7 and MMP-26) [38]. In particular, MMP-1, MMP-2, MMP-9, and MMP-13 contribute to the degradation and reorganization of the ECM components and are involved in various pathologies. A relationship between ROS generation and MMP induction has been highlighted in previous studies [39,40] as ROS can operate as second messengers, reacting with a wide range of biomolecules and inducing the cleavage of MMPs' pro-domain regions and the activation of enzymes. These findings support a mechanistic link between the free radical theory and many degenerative diseases [41]. In addition, ROS can induce the activation of key transcription factor, such as NF-κB and activator protein 1 (AP-1), whose ability to regulate MMPs' expression and activity has been previously reported [41–43].

In our study, both flavones and flavonols showed greater inhibitory effects on MMP-1 and MMP-13 than on MMP-2 and MMP-9 (see Tables 1 and 2). In particular, a decrease of antioxidant activity (ROO• and NO•) in the flavone subclass was related to a parallel decrease of their ability to inhibit MMP-1 and MMP-13 while no clear relationship was observed between the antioxidant activity and inhibition of MMP-2 and MMP-9. In the flavonol subclass, we did not observe a correlation between antioxidant activity and all types of MMPs investigated. These results suggest that, besides the number of –OH groups, the position of the hydroxyl groups may modify the interaction with the MMPs' pockets because of changes in both the molecular planarity and hydrophobicity.

On the basis of the structural heterogeneity of polyphenols together with their multiple mechanisms of action, we performed SAR research to gain a basic understanding of polyphenol activities in relation to their chemical structure.

The polyphenols are structurally characterized by the presence of one or more six-carbon aromatic rings and two or more phenolic hydroxyl groups. For flavonoids to exert their radical scavenging and/or antioxidant activity, the main structural requirements are ortho-dihydroxy structures in the B ring and a C2-C3 double bond in conjugation with a 4-oxo function in the C ring. The presence of ortho-dihydroxy structures in the B ring provides stability to the flavonoid phenoxyl radicals by expanded electron delocalization and/or hydrogen bonding ability. On the other side, a C2-C3 double bond in conjugation with a 4-oxo function in the C ring leads to the co-planarity of the hetero ring and contributes to electron delocalization over all three-ring systems, thus increasing the radical stability. Therefore, the lack of one or both the above-mentioned features makes the flavonoids less potent antioxidants [44]. In addition, the hydroxyl groups at positions 3 and 5 in the A and C ring, respectively, provide a hydrogen bonding to the oxo-group, thus increasing the radical scavenger activity [45]. Consequently, the flavone chrysin and the flavonol kaempferol were expected to show lower antioxidant activity than the other components of their subclass investigated in this work. As shown in Tables 1 and 2, the results obtained supported the validity of the above-mentioned criteria.

Both the topological polar surface area (TPSA) and the partition coefficient (log P) may play an important role in determining the interaction of flavonoids with MMPs. As shown in Table 4, the log P values of flavonols decreased as their TPSA increased (Figure 8, $r^2$ = 0.9999) while no clear relationship was observed between these two parameters in the subclass of flavones (see Table 3).

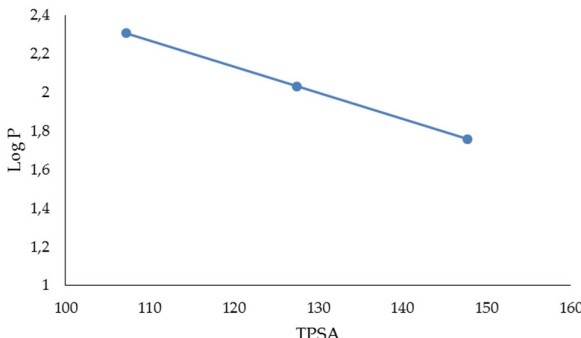

**Figure 8.** Relationships between the partition coefficient (log P) and topological polar surface area (TPSA) of flavonols.

The inhibitory effect on MMP-1 of flavones and flavonols decreased in the following order: quercetin > luteolin = myricetin > kaempferol > apigenin > chrysin. Sim et al. [28] measured the MMP1 inhibitory effects of the same flavonoids assayed in this study, reporting a 7% to 30% inhibition at 200 μM, while in the present work, a 50% inhibition was observed at < 10 μM for all investigated compounds. The discrepancy between our results and those reported by Sim et al. [28] could be attributed to the different method and experimental conditions used to determine MMP-1 inhibition. Sim et al. [28] assayed collagenase inhibition by measuring the fluorescence of collagen fragments on cleavage by MMP-1 using EnzChek Collagenase/Gelatinase kits (Molecular Probes Inc., Eugene, OR, USA). In our study, the hydrolysis inhibition of the fluorescence-quenched peptide substrate Mca-Pro-Leu-Gly-Leu-Dpa-Ala-Arg-NH2 (OmniMMP Fluorescent Substrate, Vinci-Biochem S.r.l., Firenze, Italy) due to MMPs was determined. Although these methods are conceptually similar, the reagents and experimental procedures were very different, and they could provide different results due to a different sensitivity. However, both methods provided the same trend, i.e., flavonols were stronger inhibitors than flavones, leading to the same conclusion, namely the C-3-hydroxyl group in the compounds resulted in higher inhibitory activity.

It is interesting to note that kaempferol and luteolin had equal TPSA values and similar log P while apigenin and chrysin had lower TPSA than the other flavonoids but similar log P values. These findings point at a more important role of TPSA in affecting the interaction of the investigated flavones and flavonols with MMP-1. A different trend was observed when comparing the inhibitory effect on MMP-13 of flavones and flavonols with their TPSA and log P values. All compounds under investigation showed $IC_{50}$ values ranging from 1 and 3, apart from chrysin, whose $IC_{50}$ was significantly lower ($IC_{50}$ = 10). As chrysin showed the lowest TPSA and the highest log P values among the tested compounds, these data suggest that high TPSA and low log P values could be important conditions for a good inhibitory effect on MMP-13.

As regards flavones' and flavonols' interaction with MMP-2 and MMP-9, all compounds effectively inhibited these gelatinases, but, as mentioned, the inhibitory effect was lower than that observed for MMP-1 and MMP-13 (Tables 1 and 2). Among flavones, apigenin provided the greatest inhibitory effect while quercetin was the most effective in the flavonols subclass. These results point at a lack of correlation between the physico-chemical parameters TPSA and log P and the ability of the investigated flavones and flavonols to interact with the gelatinases MMP-2 and MMP-9. Therefore, further investigations are planned to better understand the physico-chemical factors affecting flavonoids' interactions with MMP-2 and MMP-9. In addition, as flavonoid activity can benefit from their incorporation into properly designed delivery systems [46–49], in vitro studies are ongoing to evaluate the antioxidant activity and the interaction with collagenases and gelatinases of flavonoids loaded into lipid nanoparticles as carriers for skin delivery.

## 4. Conclusions

Several physico-chemical properties of flavonoids can affect their antioxidant effect, their anti-glycation activity, and their ability to inhibit metalloproteinases. The results obtained in this work support the importance of the number of hydroxyl groups in the flavone (luteolin, apigenin, and chrysin) or flavonol (mirycetin, quercetin, and kaempferol) molecules as increasing the number of -OH groups led to a proportional enhancement of the free radical scavenging ability against ROO• and NO• of the investigated compounds. When two physico-chemical parameters related to the antioxidant capacity of polyphenols, namely BDE and IP, were correlated to the observed antioxidant activity of the flavone or flavonol molecules, we found that a single parameter was not predictive of the antioxidant activity, but both parameters should be taken into account to estimate the antioxidant effect of the flavonoids under investigation.

In vitro AGE production was strongly inhibited by all flavones and flavonols tested, apart from kaempferol, thus suggesting that the presence of an hydroxyl group in the B ring was not the key factor in determining the anti-glycation activity of such molecules.

The inhibitory effect of the investigated flavones and flavonols on MMP-1 and MMP-13 could be related to their TPSA and log P values, while no clear relationship between such physico-chemical parameters and flavones' and flavonols' inhibitory activity on MMP-2 and MMP-9 was observed.

In conclusion, the results of this work suggest that the choice of suitable physicochemical parameters could allow anticipation of the in vitro antioxidant and anti-glycation activity of flavones and flavonols and their inhibitory effects on collagenases. The ability to predict flavones' and flavonols' activity from their physicochemical properties could be a valuable tool in developing novel cosmetic products for preventing and/or treating skin ageing by specifically counteracting the mechanisms involved in the oxidative and degenerative processes that lead to skin function impairment.

**Author Contributions:** Conceptualization, A.P. and S.R.; methodology, A.P. and S.R.; validation, A.P., S.R. and L.M.; formal analysis, S.R. and F.P.; investigation, F.P., G.L. and E.A.S.; writing—original draft preparation, A.P. and S.R.; writing—review and editing, A.P., S.R. and L.M.; visualization, F.P., G.L., E.A.S. and L.M.; supervision, A.P., S.R. and L.M. All authors have read and agreed to the published version of the manuscript.

**Funding:** This research received no external funding.

**Conflicts of Interest:** The authors declare no conflict of interest.

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
