# Peer review of "Natural Flavones and Flavonols: Relationships among Antioxidant Activity, Glycation, and Metalloproteinase Inhibition"

_cosmetics, doi:10.3390/cosmetics7030071_

Round 1
Reviewer 1 Report
Ronsisvalle et al. studied the effect of natural flavones and flavonols in antioxidant activity, glycation and metalloproteinase inhibition. The antioxidant activity of flavones (luteolin, apigenin and chrysin) and flavonols (mirycetin, quercetin and kaempferol) with their inhibitory effects on both metalloproteinases (MMP-1, MMP-2, MMP-9, 18 MMP-13) and advanced glycation endproducts (AGEs) formation were evaluated using binding dissociation enthalpy (BDE), ionization potential (IP), partition coefficient (Log P) and topological polar surface area (TPSA) assays. The manuscript is well written. The experiments are well designed, and data was presented well.
Some minor suggestions and questions to improve the manuscript.
In p.1; line 41-2, recent appropriate references should be cited on metalloproteinases and their inhibitors. Ref 8 is not totally relevant to the statement. “The activation of MMPs causes progressive damage to the 42 components of the extracellular matrix (ECM) [8]”.
Kaempferol is a known inhibitor of MMP-2 and MMP-9, and it was previously shown to inhibit MMP-9 (Lin et al, PLOS One, 2013; Li et al, Cell Biol, 2015), but in this study the results are contradictory. What do authors think about this inconsistency between the previous data and these results?
Do authors have any comment on the molecular mechanism’s details of how some of these compounds selectively inhibit some MMPs and not the others?
Author Response
Reviewer 1
Ronsisvalle et al. studied the effect of natural flavones and flavonols in antioxidant activity, glycation and metalloproteinase inhibition. The antioxidant activity of flavones (luteolin, apigenin and chrysin) and flavonols (mirycetin, quercetin and kaempferol) with their inhibitory effects on both metalloproteinases (MMP-1, MMP-2, MMP-9, 18 MMP-13) and advanced glycation endproducts (AGEs) formation were evaluated using binding dissociation enthalpy (BDE), ionization potential (IP), partition coefficient (Log P) and topological polar surface area (TPSA) assays. The manuscript is well written. The experiments are well designed, and data was presented well.
Some minor suggestions and questions to improve the manuscript.
Query 1.
In p.1; line 41-2, recent appropriate references should be cited on metalloproteinases and their inhibitors. Ref 8 is not totally relevant to the statement. “The activation of MMPs causes progressive damage to the components of the extracellular matrix (ECM) [8]”.
Answer
We would like to thank the reviewer for the suggestions. To comply with the reviewer’s request, we replaced references 8-10 with the following references on metalloproteinases and their inhibitors that are more recent:
[8] Raeeszadeh-Sarmazdeh, M.; Do, L.D.; Hritz, B.G. Metalloproteinases and their inhibitors: potential for the development of new therapeutics. Cells 2020, 9, 1313. DOI:10.3390/cells9051313
[9] Levin, M.; Udi, Y.; Solomonov, I.; Sagi, I. Next generation matrix metalloproteinase inhibitors — Novel strategies bring new prospects. Biochim. Biophys. Acta 2017, 1864, 1927-1939. DOI:10.1016/j.bbamcr.2017.06.009.
[10] Fields, G. B. The rebirth of matrix metalloproteinase inhibitors: moving beyond the dogma. Cells 2019, 8, 984. DOI:10.3390/cells8090984.
Query 2.
Kaempferol is a known inhibitor of MMP-2 and MMP-9, and it was previously shown to inhibit MMP-9 (Lin et al, PLOS One, 2013; Li et al, Cell Biol, 2015), but in this study the results are contradictory. What do authors think about this inconsistency between the previous data and these results?
Answer
We agree that Kaempferol is a known inhibitor of MMP-2 and MMP-9 and the results of our study pointed out an inhibitory activity higher or similar to that previously reported by others. In particular, Lin et al. reported: “MMP-2 enzyme activity was inhibited by 53% at the highest concentration of kaempferol (100 mM)” (Lin C-W, Chen P-N, Chen M-K, Yang W-E, Tang C-H, Yang S-F, et al. Kaempferol Reduces Matrix Metalloproteinase-2 Expression by Down-Regulating ERK1/2 and the Activator Protein-1 Signaling Pathways in Oral Cancer Cells. PLoS ONE 2013, 8, e80883). In addition, Li et al. highlighted that “kaempferol (20 and 40 µmol/L) strongly decreased the gelatinolytic activities of MMP-2 (reduced by 41.1% and 51.8%, respectively, n = 3, P < 0.01), and MMP-9 (reduced by 32.4% and 61.5%, respectively, n = 3, P < 0.01)” ( Li, C.; Zhao, Y.; Yang, D.; Yu, Y.; Guo, H.; Zhao, Z.; Zhang, B.; Yin, X. Inhibitory effects of kaempferol on the invasion of human breast carcinoma cells by downregulating the expression and activity of matrix metalloproteinase-9. Biochemistry and Cell Biology, 2015, 93, 16-27). In our studies, Kaempferol IC50 was 24 and 27 µM for MMP-2 and MMP-9, respectively. Therefore, our results confirmed the gelatinolytic activity of Kaempferol. However, due to the reviewer comment, we realized that we did not convey properly this concept and to highlight that all flavones and flavonols effectively inhibited MMP-2 and MMP-9 we modified the text at line 595 (revised version) as follows:
“As regards flavones and flavonols interaction with MMP-2 and MMP-9, all compounds effectively inhibited these gelatinases but, as above mentioned, the inhibitory effect was lower than that observed for MMP-1 and MMP-13 (Table 1 and 2).”
Query 3.
Do authors have any comment on the molecular mechanism’s details of how some of these compounds selectively inhibit some MMPs and not the others?
Answer
The results of our study do not allow us to offer a reliable hypothesis for the molecular mechanisms involved in the inhibition of the investigated MMPs due to the lack of correlation between the physico-chemical parameters TPSA and Log P and the ability of the investigated flavones and flavonols to interact with gelatinases MMP-2 and MMP-9. Therefore, we reported in the text at line 601 (revised version)” …., further investigations are planned to better understand the physico-chemical factors affecting flavonoids interactions with MMP-2 and MMP-9.”

Reviewer 2 Report
The paper “Natural flavones and flavonols: relationships among antioxidant activity, glycation and metalloproteinase inhibition” focuses on flavones and flavonol activity with specific regard to their antioxidant power and inhibitory effect on AGEs production, both of them useful for a potential beneficial action of such molecules against skin degeneration. The topic is interesting and falls into the Journal scopes.
Here are some points which in my opinion have to be addressed:
The Introduction should be shortened, so that the reader can more rapidly have an idea of the scope of the investigation.
The use of personal style, i.e. “we investigated”, “we evaluated”, should be limited and used only if necessary.
Flavones description should be included in the Introduction rather than in the Results-Discussion section (lines 174-200), so that the outcomes of the paper may be more promptly be discussed.
What reported in the Introduction (lines 87-89) should be much more emphasized in the concluding remarks to make the paper have more sense and be more suitable for the Journal scopes. Actually, the biological meaning of the investigation and its potential applicability in cosmetic field have been not properly considered.
After these changes, the paper can be accepted for publication on Cosmetics.
Author Response
Reviewer 2
The paper “Natural flavones and flavonols: relationships among antioxidant activity, glycation and metalloproteinase inhibition” focuses on flavones and flavonol activity with specific regard to their antioxidant power and inhibitory effect on AGEs production, both of them useful for a potential beneficial action of such molecules against skin degeneration. The topic is interesting and falls into the Journal scopes.
Here are some points which in my opinion have to be addressed:
Query 1.
The Introduction should be shortened, so that the reader can more rapidly have an idea of the scope of the investigation.
Answer
We would like to thank the reviewer for reviewing our manuscript.
We agree with the reviewer that a short introduction allows the reader to have more rapidly an idea of the aim of the work. However, as the reader is not supposed to be an expert in the investigated topic, we believe that a clear description of the background illustrating relevant works already published on this topic is essential to understand the scope of our work. As highlighted in the introduction, many other papers investigated the relationship between the antioxidant effect of several flavonoids and a single expression of their potential biological activity but none of them took into account the ability of flavonoids with different physico-chemical and antioxidant properties to affect simultaneously the glycation process and MMPs formation. Therefore, to provide the reader with basic information about the already reported correlations between physico-chemical and antioxidant properties of flavonoids, we reported some relevant papers on this topic. In our opinion, shortening the introduction would make the aim of our work not very clear.
Query 2.
The use of personal style, i.e. “we investigated”, “we evaluated”, should be limited and used only if necessary.
Answer
To comply with the reviewer’s request, we modified the personal style whenever possible.
Query 3.
Flavones description should be included in the Introduction rather than in the Results-Discussion section (lines 174-200), so that the outcomes of the paper may be more promptly be discussed.
Answer
As the reviewer pointed out in his/her first query (see query 1), the introduction should not be too long so that the reader can rapidly have an idea of the aim of the work. Therefore, to avoid lengthening the introduction, we preferred not to include the description of the chemical structures of flavones and flavonols in the introduction.
Query 4.
What reported in the Introduction (lines 87-89) should be much more emphasized in the concluding remarks to make the paper have more sense and be more suitable for the Journal scopes. Actually, the biological meaning of the investigation and its potential applicability in cosmetic field have been not properly considered.
Answer
We would like to thank the reviewer for the valuable comment. To highlight the importance of the knowledge of the relationships governing flavonoids’ activity in the cosmetic field, we added the following sentences (line 647 revised version) in the conclusions:
The ability to predict flavones and flavonols activity from their physicochemical properties could be a valuable tool in developing novel cosmetic products for preventing and/or treating skin ageing by specifically counteracting the mechanisms involved in the oxidative and degenerative processes that lead to skin functions impairment.

Reviewer 3 Report
This article entitled “Natural flavones and flavonols: relationship among antioxidant activity, glycation and metalloproteinase inhibition” by Ronsisvalle et al. investigates structure-function relation of flavonoids using 3 flavones and 3 flavonols, natural molecules that are known to have anti-oxidant activities. The values of peroxyl radical scavenger capacity, NO scavenger capacity, anti-glycation (presumably through free radical) activity are all correlated as expected. Series of structure-function studies have previously demonstrated the importance of OH groups in combination with C2-C3 double bond for the radical scavenging activity, and authors were able to successfully replicate those findings. Only the latter half of the article includes some novel information on structural features of flavonoids such as topological polar surface area, and their relations to scavenger activity, which can be useful for drug design. Part of the first half of results do not agree with the previous finding, and the authors provided no explanation.
Major Point
As the authors cited as ref#28, Sim et al. (07) measured MMP1 inhibitory effects of all 6 flavonoids used in this article. They range 7% to 30% inhibition at 200 uM, while the authors report 50% at < 10 uM for all 6 compounds. What is the source of this huge discrepancy? Before highlighting the difference between collagenases (MMP1 and MMP13) and gelatinases (MMP2 and MMP9), the authors should discuss the difference between previous study and their results.
Minor Points
L82 MMP-3 should be MMP-13. This sentence requires references.
L219 *p<0.05—which ones of the values in the table are significant? None of them has the asterisk.
L223 same as above.
L349 MPP-1 should be MMP-1.
Author Response
Reviewer 3
This article entitled “Natural flavones and flavonols: relationship among antioxidant activity, glycation and metalloproteinase inhibition” by Ronsisvalle et al. investigates structure-function relation of flavonoids using 3 flavones and 3 flavonols, natural molecules that are known to have anti-oxidant activities. The values of peroxyl radical scavenger capacity, NO scavenger capacity, anti-glycation (presumably through free radical) activity are all correlated as expected. Series of structure-function studies have previously demonstrated the importance of OH groups in combination with C2-C3 double bond for the radical scavenging activity, and authors were able to successfully replicate those findings. Only the latter half of the article includes some novel information on structural features of flavonoids such as topological polar surface area, and their relations to scavenger activity, which can be useful for drug design. Part of the first half of results do not agree with the previous finding, and the authors provided no explanation.
Major Point
As the authors cited as ref#28, Sim et al. (07) measured MMP1 inhibitory effects of all 6 flavonoids used in this article. They range 7% to 30% inhibition at 200 uM, while the authors report 50% at < 10 uM for all 6 compounds. What is the source of this huge discrepancy?
Answer
We would like to thank the reviewer for reviewing our manuscript.
The discrepancy between our results and those reported by Sim et al. could be attributed to the different method and experimental conditions used to determine MMP-1 inhibition. Sim et al. assayed collagenase inhibition measuring fluorescence of collagen fragments on cleavage by MMP-1 using EnzChek Collagenase/Gelatinase kits (Molecular Probes Inc., OR, USA.). In our study, we determined the inhibition of the hydrolysis of the fluorescence-quenched peptide substrate Mca-Pro-Leu-Gly-Leu-Dpa-Ala-Arg-NH2 (OmniMMP Fluorescent Substrate, Vinci-Biochem S.r.l., Firenze, Italy) due to MMPs. Although these methods are conceptually similar, the reagents and experimental procedures are very different and they could provide different results due to a different sensitivity. We would like to point out that both methods provided the same trend i.e. flavonols were stronger inhibitors than the flavones leading to the same conclusion, i.e. the C-3-hydroxyl group in the compounds results in higher inhibitory activity.
Before highlighting the difference between collagenases (MMP1 and MMP13) and gelatinases (MMP2 and MMP9), the authors should discuss the difference between previous study and their results.
Answer
As mentioned above, different in vitro methods could provide results that are different in absolute value. Therefore, we preferred not to compare our results with previous published ones to avoid misleading interpretations.
Minor Points
L82 MMP-3 should be MMP-13. This sentence requires references.
Answer
We apologize for this typo. We corrected the mistake and added a reference to this sentence.
L219 *p<0.05—which ones of the values in the table are significant? None of them has the asterisk.
Answer
We apologize for this typo. We deleted the sentence *p<0.05 significantly different versus control.
L223 same as above.
Answer
We apologize for this typo. We deleted the sentence *p<0.05 significantly different versus control.
L349 MPP-1 should be MMP-1.
Answer
We apologize for this typo. We corrected the mistake.

Round 2
Reviewer 3 Report
The authors gave the answer to the major point, clearly and satisfactory. This explanation should be added in L363 of the revised manuscript after the sentence that ends with “(Table 1 and 2)”. This addition is required to avoid misleading interpretations, such as intentional coverups by the authors of the inconvenient discrepancies with the previous studies. This is critical for the credibility of the journal. With this addition, the manuscript is considered to be appropriate for publication.
Author Response
Reviewer 3
The authors gave the answer to the major point, clearly and satisfactory. This explanation should be added in L363 of the revised manuscript after the sentence that ends with “(Table 1 and 2)”. This addition is required to avoid misleading interpretations, such as intentional coverups by the authors of the inconvenient discrepancies with the previous studies. This is critical for the credibility of the journal. With this addition, the manuscript is considered to be appropriate for publication.
Answer
We thank the reviewer for the suggestion. According to the reviewer’s comment, we inserted in the text the following paragraph:
Sim et al. [28] measured MMP1 inhibitory effects of the same flavonoids assayed in this study, reporting 7% to 30% inhibition at 200 µM, while in the present work 50% inhibition was observed at < 10 µM for all investigated compounds. The discrepancy between our results and those reported by Sim et al. [28] could be attributed to the different method and experimental conditions used to determine MMP-1 inhibition. Sim et al. [28] assayed collagenase inhibition measuring fluorescence of collagen fragments on cleavage by MMP-1 using EnzChek Collagenase/Gelatinase kits (Molecular Probes Inc., OR, USA.). In our study, the hydrolysis inhibition of the fluorescence-quenched peptide substrate Mca-Pro-Leu-Gly-Leu-Dpa-Ala-Arg-NH2 (OmniMMP Fluorescent Substrate, Vinci-Biochem S.r.l., Firenze, Italy) due to MMPs was determined. Although these methods are conceptually similar, the reagents and experimental procedures were very different and they could provide different results due to a different sensitivity. However, both methods provided the same trend i.e. flavonols were stronger inhibitors than flavones leading to the same conclusion, namely the C-3-hydroxyl group in the compounds resulted in higher inhibitory activity.
The reviewer suggested the insertion of the above mentioned comment at line 363 (revised version) but we preferred to insert this comment at line 351 (revised version) because at line 350-351 we illustrated the results about MMP-1 while at line 363 we discussed the results about MMP2- and MMP-9. As the comment we inserted deals with MMP-1, we believe it is more relevant for the reader to find this comment along with the results about MMP-1 rather than together the results about MMP-2 and MMP-9.
